# Chemical-Physical Behaviour of Microgels Made of Interpenetrating Polymer Networks of PNIPAM and Poly(acrylic Acid)

**DOI:** 10.3390/polym13091353

**Published:** 2021-04-21

**Authors:** Valentina Nigro, Roberta Angelini, Monica Bertoldo, Elena Buratti, Silvia Franco, Barbara Ruzicka

**Affiliations:** 1Istituto dei Sistemi Complessi del Consiglio Nazionale delle Ricerche (ISC-CNR), Sede Sapienza, 00185 Roma, Italy; valentina.nigro@enea.it (V.N.); elena.buratti@roma1.infn.it (E.B.); 2Dipartimento di Fisica, Sapienza Università, 00185 Rome, Italy; 3Dipartimento di Scienze Chimiche, Farmaceutiche ed Agrarie, Università degli Studi di Ferrara, 45121 Ferrara, Italy; monica.bertoldo@isof.cnr.it; 4Dipartimento di Scienze di Base e Applicate per l’Ingegneria (SBAI), Sapienza Università, 00185 Rome, Italy; silvia.franco@uniroma1.it

**Keywords:** microgels, soft matter, colloids, PNIPAM, poly(acrylic acid), interpenetrated, Dynamic Light Scattering, rheology, Raman spectroscopy, Small Angle Neutron Scattering

## Abstract

Microgels composed of stimuli responsive polymers have attracted worthwhile interest as model colloids for theorethical and experimental studies and for nanotechnological applications. A deep knowledge of their behaviour is fundamental for the design of new materials. Here we report the current understanding of a dual responsive microgel composed of poly(N-isopropylacrylamide) (PNIPAM), a temperature sensitive polymer, and poly(acrylic acid) (PAAc), a pH sensitive polymer, at different temperatures, PAAc contents, concentrations, solvents and pH. The combination of multiple techniques as Dynamic Light Scattering (DLS), Raman spectroscopy, Small Angle Neutron Scattering (SANS), rheology and electrophoretic measurements allow to investigate the hydrodynamic radius behaviour across the typical Volume Phase Transition (VPT), the involved molecular mechanism and the internal particle structure together with the viscoelastic properties and the role of ionic charge in the aggregation phenomena.

## 1. Introduction

Polymeric hydrogels are widely used systems for technological and biomedical applications due to their high water content and biocompatibility. Among them poly(N-isopropylacrylamide) (PNIPAM) is one of the most studied polymer for stimuli-responsive hydrogels, thanks to its very attracting temperature dependent properties. Individual chains of PNIPAM in aqueous solutions exhibit a coil to globule transition when temperature is increased above a Lower Critical Solution Temperature (LCST ≈ 305 K). Correspondingly in PNIPAM hydrogels, where the polymer chains are crosslinked, a VPT occurs at T ≈ 305 K, a temperature value interestingly compatible with physiological ones. In recent years considerable efforts have been made to find multi-stimuli-responsive systems able to enhance properties and ductility of PNIPAM hydrogels. In particular, systems with both temperature and pH sensitivity, such as PNIPAM and poly(acrilic acid) (PAAc), are of great scientific and technological importance, due to their possible use in many biomedical applications [1]. The synthesis methods to incorporate AAc into PNIPAM hydrogels, which play a crucial role in the properties of the systems, can be divided into two categories: (i) random copolymerization—PNIPAM-co-AAc—[2,3,4] and (ii) polymer interpenetration—IPN (Interpenetrated Polymer Network) PNIPAM/PAAc [5,6,7,8]. In the first case acrylic acid monomer is randomly co-polymerized with NIPAM, thus changing the temperature properties of the hydrogel. The incorporation of a large amount of acrylic acid can give rise to a reduction and even to a suppression of temperature sensitivity. On the contrary in the case of interpenetration of the two networks the hydrogel can respond independently to the two external stimuli. Finally, new approaches such as PAAc-grafted PNIPAM [1] and PNIPAM-block-PAAc [9] hydrogels with copolymer networks have been proposed in recent decades.

More recently, several scientists have proposed the use of microgels: roughly spherical particles made of crosslinked polymer networks. They have become very popular in the scientific community due to their softness originating unusual phase behaviours [10,11,12,13,14,15] and to their large variety of applications. Their submicron size leads to a faster and reversible response to external environmental stimuli with respect to macrogels making these systems particularly suitable in many applications as for example those summarized in Figure 1.

In particular, microgels are very promising as drug-delivery carriers because of their high loading capacity, high stability, and responsiveness to environmental factors, such as ionic strength, pH, and temperature that usually are not relevant parameters for common pharmaceutical nanocarriers [16,20,21,22,23,24]. They can be designed to spontaneously incorporate biologically active molecules through hydrogen bonds, electrostatic, van der Waals and/or hydrophobic interactions between the agent and the polymer matrix and provide a broad opportunities for applications requiring very small size. Such nanosized particles in the swollen state can store a large amount of drugs and release them into the target cells matching the desired kinetics of release. Due to the on demand activation by local changes in temperature or pH, microgels based on PNIPAM and PAAc gained particular relevance in anti-cancer and anti-inflammatory drug delivery, as cancer and inflammation are associated with heat generation, acidic pH and change in ionic content [25]. In this field microfluidic has been used for anchoring liposomes as extended delivery platform for treatment of osteoarthritis [17]. Monodisperse Gelatin methacryloyl-liposomes hybrid microgels have been obtained using a one-step innovational microfluidics technology. The use of this system with encapsulated kartogenin could effectively reduce osteophyte burden and prevent articular cartilage degeneration.

One of the most interesting application of these smart materials is in tissue engineering aiming at designing, building and developing living tissue models to mimic environments [18,26]. Different attempts and techniques have been explored to design scaffolds with specific physical, chemical and mechanical properties able to emulate the extracellular matrix. Polymers provide a versatile platform for this purpose and in this framework polymeric microgels have recently gained attention as a potential scaffold either alone or in combination with other scaffold for tissue engineering. To recreate the architecture of many tissues, a new approach using cell-sheet engineering with temperature-responsive culture dishes has been followed by exploiting the temperature-responsiveness of PNIPAM polymer that makes possible to control cell adhesion with simple temperature changes [18]. The addition of ionisable groups into a PNIPAM network represents a new strategy to improve the hydrogel elasticity and provide a multiresponsiveness to the system. Microgels can be used indeed as building blocks to create two-dimensional structures with high potential in various high-tech applications. Very recently different methods have been proposed to obtain water-insoluble macroscale films based on PNIPAM/AAc IPN microgels [27,28] to create thermo- and pH-sensitive films. The presence of ionizable groups, such as carboxylic acid of AAc, has been reported to affect both the packing density of the particles and their interaction with the substrate. IPN PNIPAM/PAAc microgels have been recently used as a platform for muscular cell cultures to investigate their potential as thermo and electro-responsive smart thin films [29,30]. IPN microgels represent promising candidates for substrate patterning and integration in microfabricated electronic platforms for cell culture applications.

Microgels may be also applied to obtain micro-optical devices with a wide range of applications in sensing and biosensing [19,31,32,33]. The use of microgels in sensing applications has further increased in recent years, since they are able to respond differently to analyte recognition, depending on their design and architecture. They can indeed show a wide variety of responses including expansion or contraction of the polymer network, changes in fluorescence response of a fluorophore within the microgel, changes in the diffracted wavelength in a colloidal assembly or changes in their optical properties, such as in the case of microlenses [32]. Given their inherently fast volume transition, simple fabrication techniques and the dynamic tunability of focal length, microgel-based microlens arrays are promising devices for the future development of micro-optics technologies with potential applications in optical imaging systems, telecommunications, and photolithography. In particular the incorporation of temperature and pH-sensitive monomers into the microgels, such as in PNIPAM/PAAc microgels, leads to resulting microlenses that are similarly responsive to changes in temperature and pH, allowing for their utility in sensing applications [31,32].

Moreover, in recent decades microgels have been widely used for the fabrication of photonic crystals. Concentrated suspensions of spherical colloidal particles are indeed known to order, due to repulsive interparticle interactions, into periodic arrays that reproduce the structures of atomic crystals with lattice constants that are commensurate with the optical and near-IR regions of the spectrum. As an example, colour-tunable colloidal crystals formed by the assembly of thermoresponsive PNIPAM/PAAc microgels have been designed by assembling into a close-packed colloidal crystalline array displaying Bragg diffraction that may be modulated by temperature changes [33].

Fabrication of PNIPAM-based optical devices have been recently achieved by sandwiching them between two thin Au layers [19]. These devices are of great interest for detecting solution temperature, pH, ions, glucose, and streptavidin and have been recently expanded for detection of specific DNA sequences in solution. This technology offers great promise as medical devices for diagnostics due to the ease of use, sensitivity and cheapness. Recently these devices has been investigated as humidity sensors [34]. Indeed the resultant assembly shows visual colour, and exhibits multipeak reflectance spectra whose position depends on the thickness of the microgel layer. Environmental humidity affects the layer thickness, which corresponds to changes in the device’s optical properties, thus allowing for their use as optical devices for humidity sensing.

Finally, microgels have been recently investigated for applications in cultural heritage conservation [35] taking advantage of the long tradition of macroscopic hydrogels for the cleaning of several kinds of artworks [36]. Exploiting their retentive properties, microgels are able to clean paper, ensuring a highly controlled water release from the gel matrix. Moreover, their reduced size makes microgels suitable to efficiently penetrate the porous structure of the paper and to easily adapt to the irregular surfaces of the artifacts.

The huge amount of literature demonstrates that one of the most studied responsive microgel is based on PNIPAM. Indeed PNIPAM-based microgels have been widely investigated in recent years [37,38,39,40,41,42,43,44] together with its VPT from a swollen hydrated state to a shrunken dehydrated one, as a consequence of the coil-to-globule transition of NIPAM chains [45]. This typical swelling/shrinking transition is the driving mechanism of the phase behaviour of aqueous suspensions of PNIPAM microgels [11,13,14] and can be strongly affected by concentration [10,46], solvent [47] and synthesis procedure (such as growing number of cross-linking points [48,49], different reaction pH conditions [50] or by introducing additives into the PNIPAM network [51]).

In this context, addition of poly-acrilic acid (PAAc) to PNIPAM microgel provides pH-sensitivity to the thermo-responsive microgel. As already found and discussed in the case of hydrogels, AAc can be incorporated into PNIPAM by: (i) random copolymerization or (ii) polymer interpenetration. In the case of PNIPAM-co-PAAc microgels, obtained from copolymerization of NIPAM and AAc, the temperature and pH-responsive components have significant mutual interference, the incorporation of carboxylic acid groups of AAc into PNIPAM microgel will increase the Volume Phase Transition Temperature (VPTT) and reduce the temperature responsiveness, limiting in same cases also the potential applications [52,53,54,55,56]. On the contrary, interpenetration of the PAAc and PNIPAM networks (IPN PNIPAM/PAAc microgels) [8,57,58,59,60,61,62,63,64,65], provides independent sensitivity to temperature and pH, retaining the same VPTT of pure PNIPAM microgel and allowing to make the two networks more or less dependent by changing pH.

While microgels made of PNIPAM-co-PAAc have been widely investigated both experimentally and theoretically [53,66,67,68,69], a deep investigation of PNIPAM/PAAc IPN microgels is still lacking.

Xia et al. [58] first reported the synthesis of PNIPAM/PAAc IPN microgel, through a two step polymerization method, and a comparison between the hydrodynamic and giration radii in very dilute conditions. Moreover, the group explored the drug delivery properties: the phase and viscosity behaviour for controlled drug release as a function of temperature [8,57] together with the viscoelastic behaviour and controlled release [59] were investigated. The possibility to use unique properties of these IPN microgels not only for technological applications but also to address fundamental questions such as liquid and glassy behaviour where assessed by Mattsson et al. [70] who showed how these deformable colloidal particles can exhibit the same variation in fragility as that observed in molecular liquids. A systematic characterization study on IPN particles in highly dilute conditions was performed by Liu et al. [61]. Atomic Force Microscopy (AFM) and Trasmission Electron Microscopy (TEM) were used to control particle synthesis while DLS and Infrared absorption (IR) permitted to study the VPT as a function of temperature for different pH and PAAc contents. The behaviour of the hydrodynamic radius as a function of temperature and pH has been also investigated in microcapsules with interpenetrating polymer network structure [60] and in semi-IPN nanocomposite microgels crosslinked by inorganic clay [71]. Moreover, IPN microgels have been investigated by our group through different techniques: the structural relaxation and the local structure of low concentration IPN samples at pH 5 and pH 7 have been, respectively, reported in Refs. [62,63] while the dependence on different solvents (H${}_{2}$O and D${}_{2}$O), to explore the role of H-bonds, has been reported in Ref. [65]. Moreover, in Ref. [64] the experimental radius of PNIPAM and IPN at fixed PAAc concentration has been compared with the one expected from the Flory–Rehner theory. More recently, the hydrodynamic radius of PNIPAM obtained through a new simultaneous DLS-SANS setup with DLS and Small Angle Neutron Scattering (SANS) has been compared in Ref. [72]. Finally, structural relaxation and rheological behaviour at different weight and PAAc contents have been reported in Refs. [73,74,75,76].

In this work, we review the behaviour of PNIPAM and IPN microgels as a function of temperature, concentration, PAAc content and pH that has been widely explored by combining different experimental techniques. The VPT transition has been investigated through DLS, SANS and Raman spectroscopy, in order to understand its phenomenology on a wide range of length scales and the microscopic mechanisms involved. DLS has been also largely used to explore the slowing down of the dynamics towards aggregation, moreover rheological and electrophoretic measurements have allowed to figure out the role of poly (acrylic acid), the main mechanisms behind particle aggregation and the macroscopic characteristic of the arrested states.

## 2. Experimental Methods

### 2.1. Sample Preparation

IPN colloidal gels of submicrometric size were synthesized by a two steps free radical polymerization method (Figure 2) [62]. In the first step, PNIPAM microgel is synthetized by precipitation polymerization in the presence of SDS as surfactant at a concentration below the CMC (C${}_{SDS}$ = 2.25 g/L = 7.81 mM) [62,63,73]. Then, in the second step, AAc is polymerized in the presence of the preformed PNIPAM microgel, N,N${}^{\prime }$-methylene-bis-acrylamide (BIS) as crosslinker and in the absence of any surfactant [58]. The latter step was carried out in the swollen state at T = 294 K, a temperature below the VPT, to allow AAc diffusing into the microgel and the polymer growing inside the preformed PNIPAM particles [58]. To avoid crosslinking between chains that are growing in different particles, the reaction mixture was diluted 1/10 with ultrapure water. Both reaction steps were carried out in four-necked jacketed reactors under inert atmosphere by Nitrogen. Mixtures were further deoxygenated by bubbling Nitrogen inside before reactions were started by adding the appropriate radical initiator: KPS in the first step and ammonium persulfate in the second one. N,N,N${}^{\prime },$N${}^{\prime }$-tetramethylethylenediamine (TEMED) was also added in the second step as transfer agent. IPN particles at different PAAc/PNIPAM ratio composition were prepared by stopping the second step reaction at the suitable degree of conversion of AAc. The samples were purified by dialysis (MCWO 14,000 Da) against distilled water with frequent water changes for 2 weeks after each step. The purified water dispersions were freezer-dried at 0.01 mmHg and 223 K and then dispersed in H${}_{2}$O or D${}_{2}$O at ≈5–10% wt concentration. In the case of sample in D${}_{2}$O the drying/dispersion cycle was repeated twice. Samples at different weight concentrations (%), in the following referred to as $C_{w}$, were obtained by further dilution in H${}_{2}$O or D${}_{2}$O. The chemical composition of the synthesized particles was assessed by combined ${}^{1}$H NMR spectroscopy and elemental analysis [77] and are reported in Table 1. Particles polydispersity is in the 10–15% range for PNIPAM and 15–20% for IPN microgels. Samples at pH 7.5 (in the following referred to as neutral pH) were obtained by adding NaOH solution to the samples at pH 5.5 (in the following referred to as acidic pH) obtained from the synthesis. The sample at pH 3.5 was obtained by the addition of HCl solution to the sample at pH 5.5.

### 2.2. Dynamic Light Scattering

DLS measurements were performed with a multi-angle setup in the time range between $10^{-6}$ s and 1 s. A solid state laser with wavelength of 642 nm (power of 100 mW) and single mode collecting fibers at five different scattering angles (θ = 30${}^{\circ }$, 50${}^{\circ }$, 70${}^{\circ }$, 90${}^{\circ }$, 110${}^{\circ }$) are used (Figure 3a). Time autocorrelation functions are therefore simultaneously acquired at 5 different scattering vectors *Q* = (4πn/λ) sin(θ/2) by calculating the intensity autocorrelation function $g_{2}\left(Q,t\right)=\frac{\langle I(Q,0)I(Q,t)\rangle }{{\langle I\left(Q,0\right)\rangle }^{2}}$ at different wavevectors that is well described by the Kohlrausch-Williams-Watts expression [78,79]:(1)$$g_{2}\left(Q,t\right)=1+b[{\left(e^{-{\left(t/\tau \right)}^{\beta }}\right]}^{2}$$
where *b* is the coherence factor, τ is an “effective” structural relaxation time and β describes the deviation from the simple exponential decay (β = 1), usually found in monodisperse systems, and gives a measure of the structural relaxation times distribution due to sample polydispersity. DLS measurements have been performed in the temperature range T = (293–313) K, at different PAAc content ($C_{PAAc}$ = 2.6%, 10.6%, 13.6%, 15.7% 19.2%, 24.6%) and three pH values (pH = 3.5, 5.5, 7.5) as a function of the weight concentration.

### 2.3. Small-Angle Neutron Scattering

Small-Angle Neutron Scattering measurements have been performed on the SANS2d instrument at the 10 Hz pulsed neutron source ISIS-TS2. The Q-range from 0.04 to 7 nm${}^{-1}$, corresponding to length scale from 0.1 to 160 nm, allows to explore the local structure of the microgel particles when they go from the fully swollen to the completely shrunken state [80,81,82]. The SANS scattered intensity $I\left(Q\right)$ is:(2)$$I\left(Q\right)={\left(\Delta \rho \right)}^{2}NV_{polymer}^{2}P\left(Q\right)S\left(Q\right)$$
where *Q* is the scattering vector with n = 1 for neutrons, Δρ is the contrast factor between the polymer and the surrounding solvent; *N* is the number density of the particles; $V_{polymer}$ is the volume of polymer within the particle; P(Q) is the form factor and S(Q) is the static structure factor. In dilute suspensions the problem is reduced to model the form factor P(Q) that in the case of crosslinked polymeric particles is not trivial. According to Shibayama et al. [81,82,83], similar network systems can be modeled through a deformable lattice model of blobs with two characteristic length scales: a short correlation length, ξ, for the rapid fluctuations of the polymer chain position, and a long correlation length, $R_{g}$, associated to the regions with higher polymer density and arising from the constraints imposed by junction points or clusters of such points (blobs). The scattered intensity can be therefore modelled as:(3)$$I\left(Q\right)=\frac{I_{L}\left(0\right)}{{\left\{1+\left[\left(D+1\right)/3\right]\xi^{2}Q^{2}\right\}}^{D/2}}+I_{G}\left(0\right)exp\left(-R_{g}^{2}Q^{2}/3\right)$$
where $I_{L}\left(0\right)$ and $I_{G}\left(0\right)$ are scale factors dependent on the polymer-solvent contrast and on the volume fraction of the microgel, ξ is the correlation length related to the size of the polymer network mesh, *D* is the Porod exponent, giving an estimate of the roughness of the interfaces between different domains of inhomogeneities and $R_{g}$ can be interpreted as the mean size of the static inhomogeneities introduced by the chemical cross-links. SANS measurements have been performed in the temperature range T = (299–315) K on D${}_{2}$O suspensions at different concentrations ($C_{w}$ = 0.1%, 0.2% and 0.3%), at $C_{PAAc}$ = 13.6% and at acidic and neutral pH.

### 2.4. Raman Spectroscopy

Raman measurements have been performed through a Horiba HR-Evolution microspectrometer in backscattering geometry, equipped with a He-Ne laser, λ= 632.8 nm and 30 mW output power (∼15 mW at the sample surface). A state-of-the-art optical filtering device based on three BragGrate notch filters [84] allows to remove the elastically scattered light and to collect Raman spectra at very low frequencies, down to 10 cm${}^{-1}$ from the laser line. The detector was a Peltier-cooled charge-coupled device (CCD) and a 600 grooves/mm grating with 800 mm focal length allows a resolution better than 3 cm${}^{-1}$. The spectrometer was coupled with a confocal microscope supplied with a set of interchangeable objectives with long working distances and different magnifications. A 20 × –0.35 NA objective has been used for the reported experiment. Measurements have been performed on aqueous suspensions of PNIPAM and IPN microgels at fixed PAAc concentration ($C_{PAAc}$ = 19.2%) in the temperature range T = (293–313) K across the VPT, at different weight concentrations and acidic pH.

### 2.5. Rheological Measurements

The viscoelastic properties of IPN microgels were probed using a rotational rheometer Anton Paar MCR102 with a cone-plate geometry. The plate diameter was d = 49.97 mm, the cone angle β = 2.006${}^{\circ }$ and the truncation of 212 μm [75]. Temperature was controlled by a Peltier system in the bottom plate connected to a water bath. To prevent sample evaporation the plates were covered with a rheometer integrated hood system designed for this aim. To probe the linear response, rheological measurements were carried out within the linear viscoelastic region that is achieved at sufficiently small values of the applied strain (γ) when the loss (G${}^{\prime \prime }\left(\omega \right)$) and storage (G${}^{\prime }\left(\omega \right)$) moduli are not strain dependent. Frequency sweeps were performed in the range f = (10${}^{-2}$–10) Hz with ω = 2πf. Viscosity instead was probed through steady shear measurements. The viscosity behaviour of IPN microgels across the VPT has been investigated at $C_{PAAc}$ = 15.7% and $C_{w}$ = 0.3% in the temperature range T = (293–313) K (Figure 7c), the loss and storage moduli have been measured at $C_{PAAc}$ = 24.6%, $C_{w}$ = 0.4%, 0.9%, 3.6% and T = 311 K at pH 5.5 (Figure 15).

### 2.6. Electrophoretic Measurements

Electrophoretic mobility of microgel suspensions has been measured by means of a MALVERN NanoZetasizer apparatus equipped with a 5 mW HeNe laser (Malvern Instruments LTD, Worcestershire, UK). This instrument employs traditional Laser Doppler Velocimetry (LDV) implemented with Phase Analysis Light Scattering (PALS) for a more sensitive detection of the Doppler shift [85]. LVD measurements are performed using the patented “mixed mode” measurement M3 where both a fast field (FF) and a slow field (SF) are applied. In FFR the field is reversed 25–50 times per s, thus making electro-osmosis insignificant and providing accurate mean mobility value. The SFR contributes extra resolution for a better distribution analysis [86,87]. The frequency shift Δν due to the mobility μ of the scattered particles under the action of the applied field E is measured by comparing the phase Φ of the scattered signal to that of a reference one, since Φ=ν· time. The mobility μ=V/E is then calculated from the relation Δν=2Vsin(θ/2)/λ) with *V* the particle velocity, θ the scattering angle and λ the laser wavelength. By a preliminary conductivity measurement, the instrument establishes a suitable electric field for a good mobility detection. Both PNIPAM and IPN samples at the different PAAc contents have been measured at $C_{w}$ = 0.05% and acidic pH. Measurements have been performed by using the dedicated U-cuvette DTS1070, in a thermostated cell by performing a ramp from 293 to 316 K with temperature step of 1 K and a thermalization time of 300 s at each step.

## 3. Results

### 3.1. Particle Size and Volume Phase Transition

The IPN samples at different PAAc concentrations were characterized from their hydrodynamic radius R${}_{H}$ obtained through DLS measurements in the high dilution limit. The values were determined from structural relaxation time τ of the intensity correlation functions g${}_{2}\left(Q,t\right)$ (Figure 3b, Equation (Equation 1)) using the Stokes Einstein relation $R_{H}$ = $K_{B}$T/6πηD, where D is the translational diffusion coefficient obtained through the relation τ = 1/Q${}^{2}$D and η is the sample viscosity that has been approximated with the solvent one. The temperature behaviour of hydrodynamic radius allows to investigate the dependance of the VPT on several parameters such as PAAc content, pH and solvent. The VPTT of IPN microgels in water without pH corrections is almost the same of the net PNIPAM regardless on the PAAc composition, as expected for IPN microgels. In fact the absence of chemical bonds between PAAc and PNIPAM guarantees that the temperature responsivity of the PNIPAM network is not changed by the PAAc network. This is clearly evident from Figure 4a and Figure 5 where the temperature behaviour of hydrodynamic radii and diameters for PNIPAM ad IPN microgels as reported, respectively, by Refs. [59,61] are shown. However, the swelling capability at the swollen/shrunken transition is highly affected by the acrylic acid content, being strongly reduced at increasing C${}_{PAAc}$. In fact, both in Figure 4a and Figure 5b the particle dimensions are increased but the swelling capability is reduced by increasing poly acrilic acid contents. This is explained by the increase of topological constraints due to the PNIPAM and PAAc networks interpenetration and to the ionic contribution to the osmotic pressure due to the presence of the PAAc. The behaviour of the hydrodynamic radius at T = 293 K (below the VPTT) and T = 313 K (above the VPTT) as a function of the PAAc content is shown in Figure 4b. PNIPAM particles (corresponding to C${}_{PAAc}$ = 0) from the first synthesis step have a size in the nanometer range, the interpenetration of 2.6% of PAAc into the PNIPAM microgel to obtain IPNs does not result in a detectable change in the particle size, indicating that this PAAc amount was well incorporated inside the PNIPAM particles, in good agreement with previous findings [77]. On the other hand, as the amount of PAAc is increased above C${}_{PAAc}$ = 2.6% an almost linear increase of the particle size is observed. For these high diluted samples TEM images as the one shown in Figure 4c were obtained. In all these cases particles look like isolated spheres, only in the case of the highest PAAc/PNIPAM ratio few twin particles are observed by TEM. The presence of permanent bonds between the twin particles is not proved.

The interpenetration of PAAc network strongly affects the interaction between PNIPAM and water molecules. Indeed, the presence of PAAc into the IPN networks provide the systems with an additional control parameter: at acid pH IPN particles exhibit a smaller hydrodynamic diameter and a smoother transition than at neutral condition. This is clearly shown in Figure 6 where the dependence on pH and solvent conditions are shown. The reduction of the particle size at acidic pH (Figure 6a) suggests that the different degree of dissociation of the COOH groups of PAAc favours hydrogen bonds with the amide groups of PNIPAM resulting in a higher shrinkage of the microgel particles [61]. The behaviour of the hydrodynamic radius with pH has been investigated in Refs. [58,61]. The role of the polymer/solvent interaction is further demonstrated considering H/D isotopic substitution in the solvent [88], as shown in Figure 6b where R${}_{H}$(T) is reported in the case of D${}_{2}$O suspensions. A similar trend is observed with a small increase of the hydrodynamic radius in neutral pH condition. Therefore, the balance between polymer/polymer and polymer/solvent interactions strictly depends on the solvent. It is worth noting that eventual differences among samples reported in the paper can be attributed to slight differences in the pH condition in the range of between pH 4.5 and 5.5.

We can thus conclude that interpenetration of PAAc network introduces additional control parameters for the interaction between PNIPAM and water molecules: polymer/solvent interaction can be therefore controlled through PAAc content [73], by varying pH or by changing solvent [64].

The microscopic volume phase transition of particles is accompanied by the change of a macroscopic quantity, the viscosity η, reported in Figure 7c as a function of temperature for an IPN microgel at $C_{PAAc}=24.6\%$, C${}_{w}$ = 0.3 %, pH 5.5 and $\dot{\gamma }=1s^{-1}$. A slow decrease is observed at low temperature, $\eta (\dot{\gamma })$ until an abrupt transition take place followed by a decrease further increasing T. The viscosity rise on heating is a counterintuitive phenomenon, observed also in systems characterized by inverse melting [89,90], that in the case of PNIPAM based microgels can be attributed to inter-chain interactions induced by shear that promote particle aggregation [91].

### 3.2. Molecular Mechanism Driving the VPT

The shrinking of PNIPAM-based microgels across the VPTT has been reported to involve molecular changes that can be probed by Raman spectroscopy [92]. Raman spectra of both PNIPAM and IPN microgels (Figure 8a) are indeed dominated by the contributions associated to C-C and C-H vibrational modes, mainly derived from NIPAM [93]. In particular the stretching bands in the 2850 and 3000 cm${}^{-1}$ spectral region, ascribed to vibrations of the CH${}_{2}$ and CH${}_{3}$ groups in the isopropyl moiety (Figure 2a), have been reported to be sensitive to hydrogen bond variations involving the amide functionality. In order to deep inside the interaction mechanism driving the VPT of PNIPAM and IPN microgels, the four superimposed bands in the indicated spectral region have been deconvoluted with four Gaussian contributions (Figure 8b). These contributions have been assigned to the following mode of PNIPAM in the hydrated state (T = 297 K) [94,95]: symmetric stretching of CH${}_{3}$ (2880 cm${}^{-1}$), symmetric and antisymmetric stretching of CH${}_{2}$ (2920 cm${}^{-1}$ and 2945 cm${}^{-1}$, respectively), antisymmetric stretching of CH${}_{3}$ (2988 cm${}^{-1}$). The intensity ratio between the symmetric and antisymmetric CH${}_{2}$ stretching modes of the methylene group (lower and higher frequency central peaks, respectively), is usually related to the lateral packing density of the polymer chains [96] and any changes can be exploited to study variation in the polymer/polymer and polymer/solvent interactions. In particular, the observed increase of the intensity ratio across the VPTT in IPN microgels suggests that the coil-to-globule transition of PNIPAM induces an increase of the packing density in the shrunken state even when PAAc is interpenetrated within the PNIPAM network.

Moreover, the highest number of water molecules surrounding the CH${}_{3}$ groups correlates with the highest frequency of the antisymmetric CH${}_{3}$ stretching (peak at 2988 cm${}^{-1}$) [97,98]. Therefore, the frequency red-shift with temperature of the CH${}_{3}$ stretching vibration (Figure 8b) is the main evidence of the dehydration of the isopropyl group of NIPAM (Figure 2). The reorganization of the neighbouring water molecules leads to the dehydration of the isopropyl group in both PNIPAM and IPN microgels, as shown by the decrease with temperature of the frequency peak and the sharp transition at the VPTT (Figure 9). The temperature behaviour of the CH${}_{3}$ frequency for IPN microgels confirms that the main features of the coil-to-globule transition of PNIPAM are preserved, although the reduced hydration of the isopropyl group of NIPAM (first drop) is accompanied by a new mechanism (evidenced by the additional bump at the VPTT) due to the steric hindrance of PAAc limiting the microgel shrinking. The combined effect of reduced hydration of the isopropyl group of PNIPAM and of the topological rearrangements of the polymer networks within the microgel particle lead to interesting differences between PNIPAM and IPN microgels. Their hydrophobicity is enhanced if PNIPAM and PAAc networks are interpenetrated perturbing the role played by water molecules. At this pH, the intra-particle and inter-particle interactions between CONH (PNIPAM) and COOH (PAAc) groups make, respectively, IPN microgel more hydrophobic and favour aggregation.

### 3.3. Local Structure across the VPT

The local structure response of microgel particles across the VPT has been investigated by Small-Angle Neutron Scattering. In Figure 10a, the scattered intensity for IPN microgels in D${}_{2}$O at fixed weight concentration $C_{w}$ = 0.1% and acidic pH are reported at two different temperatures (below and above the VPTT). The deformable lattice model of blobs, introduced for PNIPAM [81,82] (Equation (Equation 3)), accurately fits our data.

The Porod exponent reported in Figure 10b, which gives an estimation of the roughness of the domain interfaces, increases above the VPTT, indicating the formation of smoother interfaces between different domains. The local intra-particle structural response to temperature across the VPT has been rationalized by looking at the behaviour of the correlation lengths ξ and of the mean size of the inhomogeneities domains $R_{g}$, reported in Figure 11a at fixed concentration and pH, as an example. The correlation length values ξ are much smaller with respect to particle dimensions (Figure 6b) and decrease above the VPTT, when microgel particles collapse and the lost of individuality of the frozen blobs gives rise to a large cluster of cross-linked points of size $R_{g}$. Therefore, when the system undergoes the macroscopic transition from the swollen to the shrunken state, the local structure experiences a transition from an inhomogeneous structure, where the lattice is deformed and the open network can accommodate large amount of water, to a porous solid-like structure, where a unique larger sized cluster is formed and the nanometric structure of the tridimensional network is lost, due to the shrinking of the polymer chains along with water expulsion (see Figure 11b).

### 3.4. Concentration Dependence

The temperature behaviour of the structural relaxation time τ and the shape parameter β have been obtained by fitting the $g_{2}\left(Q,t\right)$ from DLS measurements with Equation (Equation 1), for both PNIPAM and IPN microgels at four weight concentrations (C${}_{w}$ = 0.1%, C${}_{w}$ = 0.3%, C${}_{w}$ = 0.5% and C${}_{w}$ = 0.8%) (Figure 12). For pure PNIPAM microgels the dynamical transition associated to the VPT is observed in the whole concentration range: the structural relaxation time τ slightly decreases with increasing temperature up to the VPTT, where the fastening of the dynamics related to the reduced size of particles is observed (Figure 12a) [62,64]. In the case of IPN microgels a more fascinating scenario shows up. At the lowest investigated concentration ($C_{w}=0.1\%$) the structural relaxation time τ slightly decreases as temperature increases up to the VPTT where a transition to its lowest value takes place as in the case of pure PNIPAM microgels. However, as $C_{w}$ increases the structural relaxation time above the VPTT suddenly grows up, indicating a slowing down of the dynamics and the formation of aggregates. This behaviour is more pronounced higher is the sample concentration and is accompanied by a viscosity increase, which is clearly observed by eyes. The dynamical transition associated to the VPT is also observed in the discontinuous temperature behaviour of the β parameter that gives a measure of the structural relaxation times distribution and is therefore the most reliable marker for sample polydispersity. This can be an intrinsic polydispersity, due to the synthesis procedure and therefore variable from sample to sample but can also be an “apparent” polydispersity that may be explained in terms of fluctuations of the particle shape at different temperatures and weight concentrations or as a result of particle aggregation (Figure 12b). In PNIPAM microgels the sample polydispersity increases with increasing $C_{w}$, as indicated by the concentration dependence of the β parameter at temperatures below the VPTT. Above the VPTT no differences depending on $C_{w}$ are observed suggesting that PNIPAM microgels in the shrunken state have low polidispersity regardless to their concentrations. A similar behaviour is also found for IPN microgels even if the more complex inner structure of IPN particles reflects in a higher polydispersity and lower β values than those for PNIPAM microgels. Intriguing differences on concentration are also observed: for C${}_{w}<$ 0.3% β decreases upon crossing the VPTT, while for C${}_{w}\geq$ 0.3% it increases with a more and more pronounced jump across the VPT with increasing concentration.

As a result both τ and β sign the existence of a crossover concentration (C${}_{w}$ = 0.3% in IPN microgels at $C_{PAAc}$ = 19.2%) above which interparticle interactions become important, giving rise to aggregation. Above the VPTT, due to the reduced particle size, Van der Waals attraction becomes stronger, thus affecting microgel aggregation. If particles are charged, as in the case of IPN microgels, also electrostatic interactions have to be taken into account, expected to be much stronger the higher the PAAc content is [73]. The presence of PAAc affects the balance between hydrophobic and hydrophilic interactions. At this PAAc content ($C_{PAAc}$ = 19.2%) and acidic pH (pH 5.5), where the fraction of deprotonated AAc moieties (COO${}^{-}$) is small but not negligible, the collapse of the PNIPAM network above the VPTT is supposed to favour the exposure of PAAc dangling chains and consequently interparticle interactions and aggregates formation. Charge density is indeed a crucial parameter for understanding IPN microgels behaviour. Electrophoretic measurements on IPN microgels have been thus performed and compared with those on PNIPAM microgels. For both PNIPAM and IPN microgels the mobility μ is affected by the volume phase transition with a decrease across the VPT (Figure 13). Interestingly, the electrokinetic transition temperature T${}_{0}$ is higher than the VPTT, as previously shown for PNIPAM-based microgels [99,100,101]. The effective charge carriers are indeed mainly confined to the outer shell that fully collapses when the VPT is totally crossed, thus shifting forward the transition temperature.

As a matter of fact, when interparticle interactions are not negligible, aggregation is driven by temperature, as a result of the microgel shrinking. To explore the slowing down of the dynamics across the VPT, we can look at the concentration behaviour of the structural relaxation time (Figure 14a). Below the VPTT, data are well fitted through an Arrhenius behaviour:(4)$$\tau =\tau_{0}\;exp\left(\;AC_{w}\right)$$
where *A* is a constant and $\tau_{0}$ is the characteristic structural relaxation time for low $C_{w}$ values. Above the VPTT, the concentration dependence of the structural relaxation time is well described by a super–Arrhenius behaviour, usually modelled by the Vogel–Fulcher–Tammann (VFT) equation:(5)$$\tau =\tau_{0}\;exp\left(\;\frac{D_{C_{w}}C_{w}}{C_{w0}-C_{w}}\right)$$
where $C_{w0}$ sets the apparent divergence, $D_{C_{w}}$ controls the growth of the structural relaxation time and $\tau_{0}$ is the characteristic structural relaxation time in the high dilution limit. The sharper concentration dependence of the structural relaxation time at high temperature can be related to the increased interactions between different particles by the COOH groups of PAAc dangling chains that are much more exposed in the shrunken state than in the swollen one. The existence of two different behaviours below and above the VPTT are also confirmed by the shape parameter β (Figure 14b), showing a minimimum in the collapsed state at a critical concentration value, as reported in Ref. [74].

The existence of different regimes is also evidenced by the behaviour of the storage $G^{\prime }$ and loss moduli G″ vs. frequency f reported in Figure 15 above the VPT for three different concentrations. In Figure 15a, $G^{\prime }$ and G″ vs. f for a sample at C${}_{PAAc}$ = 24.6% and T = 311 K are shown. At C${}_{w}$ = 0.4% the power low behaviour of the moduli is typical of the liquid state, as C${}_{w}$ is increased, at C${}_{w}$ = 0.9%, above the glass transition concentration found by the VFT model for the structural relaxation time, $G^{\prime }$ > $G^{\prime \prime }$ over the entire frequency range indicating a viscoelastic behaviour typical of a solid like system. For this concentration the plateau modulus is not perfectly constant indicating a weak solid-like behaviour as deeply explained in Ref. [76]. A further increase of moduli is observed at C${}_{w}$ = 3.6%, they are frequency independentand and $G^{\prime }>>G^{\prime \prime }$ revealing a solid like response of the system that indicates the existence of another state [76]. These results are in good agreement with measurements from Ref. [59] at T = 310 K reported in Figure 15b where the open and closed symbols for $G^{\prime }$ and $G^{\prime \prime }$ are inverted with respect to Figure 15a.

In order to understand the PAAc role on the aggregation process across the VPT, the temperature behaviour of the structural relaxation time and the β parameter at different PAAc content has been explored through DLS. In Figure 16a, the well known dynamical transition associated to the particle shrinking is evidenced for both PNIPAM and IPN microgels [62,64] together with two opposite temperature behaviours above the VPTT: τ(T) decreases for PNIPAM and IPN at the lowest investigated PAAc content while it increases for IPN samples at higher $C_{PAAc}$, with a very significant growth at the highest $C_{PAAc}$. Furthermore, the β parameter, reported in Figure 16b, clearly points out the strong influence of the PAAc concentration. Three different trends can be observed indeed, one for PNIPAM and IPN at the lowest $C_{PAAc}$, one for IPN with intermediate $C_{PAAc}$ and the latter for IPN at the highest $C_{PAAc}$ = 24.6%. PAAc therefore represents a good experimental control parameter for tuning inter-particle interactions and aggregation: the higher the amount of acrylic acid interpenetrating the PNIPAM network, the more the aggregation and the viscosity increase are favoured.

To highlight these dynamical changes related to the interpenetration of the poly (acrylic acid) within the PNIPAM network, one can compare the behaviours of the structural relaxation time τ and the shape parameter β as a function of PAAc content, at temperature below and above the VPTT (Figure 17a,b). The jump in a range of $C_{PAAc}$ = (2.6 ÷ 10.6)% points out a cross-over between two different regions, with a critical value of PAAc content expected in the range $C_{PAAc}^{*}$ = (7 ÷ 8)%: below this $C_{PAAc}^{*}$ IPN microgels at pH 5.5 behave very similarly to pure PNIPAM microgel, indicating that the charges influence is negligible, while above $C_{PAAc}^{*}$ the effect of PAAc, and therefore of charge density, becomes relevant, leading to a slowing down of the dynamics (increase of the structural relaxation time) and an enhancement of polydispersity (decrease of the β parameter). At even higher PAAc content the structural relaxation time suddenly grows up as a consequence of the larger aggregates formation eventually clustering in a percolating network while the β parameter is not changing significantly. This scenario is maintained above the VPTT where however the β parameter is increased, indicating that in the collapsed state the sample polidispersity decreases, as previously discussed.

To better emphasize how the PAAc content affects the dynamics of the system, the normalised structural relaxation time and the β parameter at temperature above the VPTT (T = 311 K) are reported in Figure 18 as a function of weight concentration $C_{w}$ at different PAAc contents. The higher is the PAAc content the stronger is the concentration dependence of the normalised structural relaxation time with divergence at high concentration (Figure 18a). The effect of PAAc on the β parameter is even more dramatic (Figure 18b), with the appearance of a minimum connected with the emerging of distinct anomalous mechanisms for particle motion [74]. These results point out that, by changing PAAc content, different dynamical behaviours can be achieved in microgels due to the increasing interactions of COOH groups belonging to PAAc chains [73].

The role of the charge density across the VPT can be explored through the temperature behaviour of the electrophoretic mobility for PNIPAM and IPN microgels. Interestingly, the magnitude of the mobility variation depends on $C_{PAAc}$ and is more pronounced at high PAAc content (Figure 19a), where collapsed IPN microgels are characterized by more negative mobility values, as expected from the increase of the charge density. For pure PNIPAM microgels the very low mobility below the VPTT reflects the low charge density, derived by the ionic initiator (KPS). Since the negative electrical charges brought by the anionic sulfate groups are covalently bonded, the total charge per particle is constant and the charge density increases upon shrinking [99,100]. For IPN microgels a similar mechanism still holds for the temperature behaviour of the electrophoretic mobility. Due to additional charged groups belonging to AAc moieties, more negative values are found for $C_{PAAc}$ > 2.6%. Moreover, the increase of the charge density above the VPTT because of particle shrinking leads to more negative mobility values for IPN microgels. These results are emphasized in Figure 19b where the mobility is shown as a function of the PAAc content at temperature below and above the VPTT. In agreement with findings from DLS measurements (Figure 17) the mobility strongly depends on PAAc content above a critical value $C_{PAAc}^{*}$ and is significanly enhanced in the collapsed state for T > VPTT. These results further support the possibility to control the effective charge density on the microgel surfaces through the amount of poly (acrylic acid) interpenetrating the PNIPAM network and hence also inter-particle interactions and aggregation phenomena.

### 3.5. pH Dependence

In the previous subsection, it has been shown how the behaviour of IPN microgels depends on PAAc content. However, since PAAc is strongly pH-responsive we can expect that IPN microgels are also strongly affected by pH variations. As an example the temperature behaviour of the local structure (discussed in Section 3.3) shows interesting differences at acidic and neutral pH. The behaviour of the Porod exponent D and of the correlation length ξ as a function of temperature (Figure 20a,b, respectively) highlights how the transition from an inhomogeneous to a porous solid-like structure across the VPT depends on pH. Moreover, the gyration radius $R_{g}$ (Figure 20c) increases with temperature in both acidic and neutral pH conditions, but at pH 7 an evident discontinuity shows up at T≈305 K, around the expected VPTT. On the contrary at pH 5 the differences between inhomogeneities domains are partially lost and the temperature responsiveness of the system is limited by the presence of the PAAc chains. The sharper transition observed at neutral pH with respect to that at acidic pH is related to the solvation in water of the PAAc chains at pH values above 5.5, due to its deprotonation. Indeed at low pH the PAAc chains are not effectively solvated by water, thus the formation of H-bonds between PAAc and PNIPAM is favoured and introduces spatial constraints which limit the PNIPAM network swelling. At neutral pH the deprotonation of the PAAc chains results in their effective hydration and the two networks are independent, with a complete regain of the swelling capability of the system.

The temperature behaviour of the structural relaxation time and β parameter as obtained from DLS for IPN microgels at C${}_{PAAc}$ = 19.2%, $C_{w}$ = 0.8% and at pH 3.5, pH 5.5 and pH 7.5 are reported in Figure 21. The huge growth of τ(T) (Figure 21a) above the VPTT at pH 7.5, indicates a serious slow down of the dynamics fairly due the formation of large aggregates. At this pH, the COOH carboxylic groups of PAAc (Figure 2) are dissociated into COO${}^{-}$, thus H-bonding between COOH groups of AAc moieties belonging to different particles are not favoured. As a consequence, the aggregation can be mainly ascribed to like-charge attraction since at this pH IPNs behave as polyelectrolyte microgels where attraction can be interpreted as a result of counterion fluctuation due to the formation of temporary dipoles [102]. At an intermediate pH of 5.5, there is a fraction of COOH groups and a not negligible fraction of COO${}^{-}$ groups, therefore aggregation can be described as a combination of both like-charge attraction and H-bonding interaction between COOH groups. Finally, at pH 3.5 the COOH groups of PAAc are fully protonated (neutralized), electrostatic interactions are excluded and H-bondings with the amidic (CONH) groups of PNIPAM (Figure 2) inside the particles are largely favoured. Nevertheless small aggregates above the VPTT are formed, suggesting that inter-particle interactions at high C${}_{PAAc}$ are not excluded and can be mainly ascribed to strong H-bonding and hydrophobic interactions. A strong pH dependence is also shown by the β parameter (Figure 21b) exhibiting similar shapes for pH 3.5 and pH 5.5 and a non trivial behaviour with a maximum at the VPTT for pH 7.5. Despite a few speculations may be proposed, this interesting behaviour needs further investigations.

## 4. Conclusions

Temperature and pH responsive microgel composed of interpenetrating polymer networks of PNIPAM and poly(acrylic acid) were synthesized through a free radical polymerization method. Microgel size was determined through DLS while the internal structure was investigated through SANS that, combined with Raman spectroscopy, allowed to determine the microscopic mechanisms driving the VPT typical of this system. The study was performed as a function of temperature, concentration, PAAc content and pH. Temperature induces a VPT of microgel particles from a swollen hydrated state to a shrunken dehydrate one. Increasing concentration induces a slowing down of dynamics and the possibility to reach arrested states whose nature is beeing explored. Synthesizing particles with different contents of poly (acrylic acid) allows to span from behaviour very similarly to those of pure PNIPAM, at low PAAc contents, to significantly different ones. This is related to an increase of the effective charge density due to the COOH groups of PAAc chains, as shown by electrophoretic mobility measurements. With increasing PAAc content, in fact, attractive interactions between protonated COOH and deprotonated COO${}^{-}$ groups belonging to different particles are enhanced and aggregation processes favoured as also evidenced by rheological measurements. Finally, a fine control of pH allows to manipulate the COOH groups of PAAc: when pH is lower than pKa (4.5 for PAAc), hydrogen bonds between these groups and the amide groups of the PNIPAM component are favoured with a consequent shrinkage of the microgels. As pH increases above pKa, the –COOH groups are gradually dissociated into COO${}^{-}$ groups, leading to swelling of the microgels. Therefore, PAAc and pH effect are strictly interconnected, with increasing PAAc content, the pH responsiveness of IPN microgels is enhanced.

## Figures and Tables

**Figure 1 polymers-13-01353-f001:**
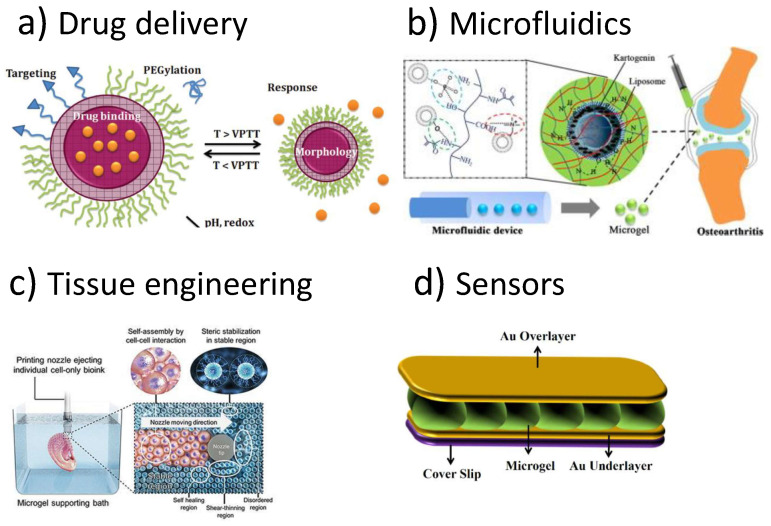
Some examples of technological applications of microgels: (**a**) as delivery vectors using functionalized PNIPAM microgels with different functional group distributions (figure adapted with permission from Ref. [16]); (**b**) in microfluidics to anchor liposomes for treatment of osteoarthritis (figure adapted with permission from Ref. [17]); (**c**) in tissue engineering using PNIPAM (figure adapted with permission from Ref. [18]); (**d**) as sensors based on PNIPAM and AAc microgels (figure adapted with permission from Ref. [19]).

**Figure 2 polymers-13-01353-f002:**
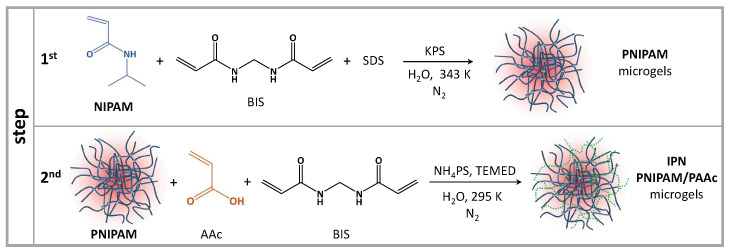
Sketch of the two steps free radical polymerization method for synthesis of IPN microgels.

**Figure 3 polymers-13-01353-f003:**
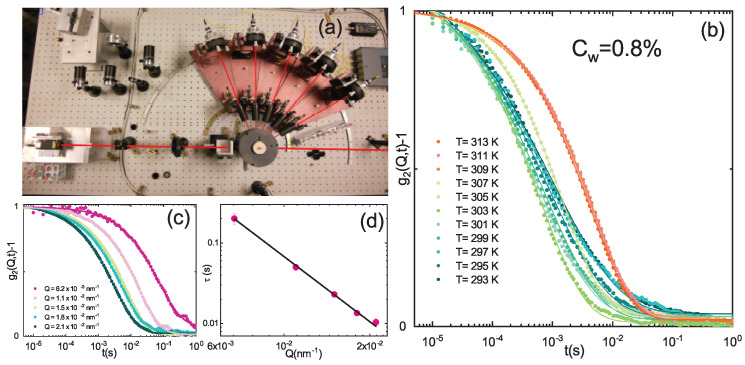
(**a**) Multiangle DLS set-up. Normalized intensity autocorrelation functions for IPN microgels at C${}_{w}$ = 0.8%, C${}_{PAAc}$ = 24.6% and pH 5.5 (**b**) as a function of temperature, at Q = 0.018 nm${}^{-1}$ and (**c**) as a function of scattering vector Q at T = 311 K. Solid lines are fits according to Equation (Equation 1). (**d**) Structural relaxation time as a function of the scattering vector.

**Figure 4 polymers-13-01353-f004:**
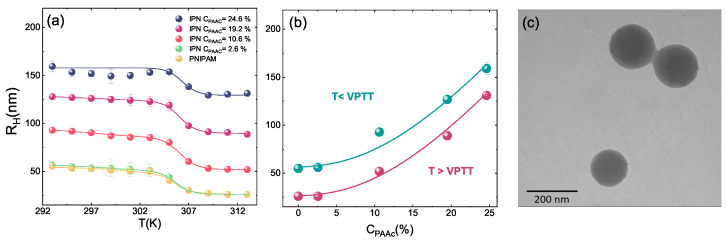
Behaviour of hydrodynamic radii from DLS measurements for IPN microgels at pH 5.5 and $C_{w}=0.01$% (**a**) as a function of Temperature (**b**) as a function of PAAc content at T = 293 K (below the VPTT) and at T = 313 K (above the VPTT) and (**c**) TEM image of IPN microgels at low PAAc concentration.

**Figure 5 polymers-13-01353-f005:**
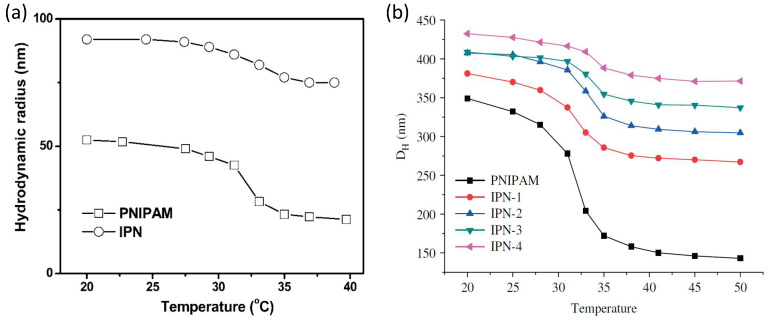
Temperature dependence for diluted PNIPAM and IPN microgel dispersions at pH = 7.0 of (**a**) hydrodynamic radii and (**b**) hydrodynamic diameters at difference PAAc contents (increasing from IPN-1 to IPN-4). Figures adapted with permission from Refs. [59,61], respectively. Ref. [59], Copyright 2021 American Chemical Society.

**Figure 6 polymers-13-01353-f006:**
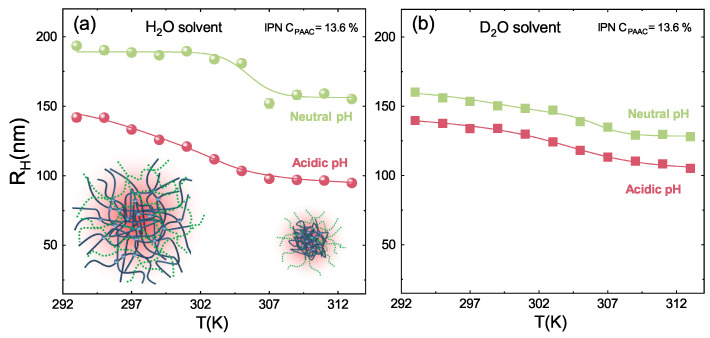
Temperature dependence of the hydrodynamic radii from DLS measurements for IPN microgels at $C_{w}=0.1$ % and C${}_{PAAc}$ = 13.6% at acidic and neutral pH in (**a**) H${}_{2}$O solvent and (**b**) D${}_{2}$O solvent. Solid lines are guide to eyes.

**Figure 7 polymers-13-01353-f007:**
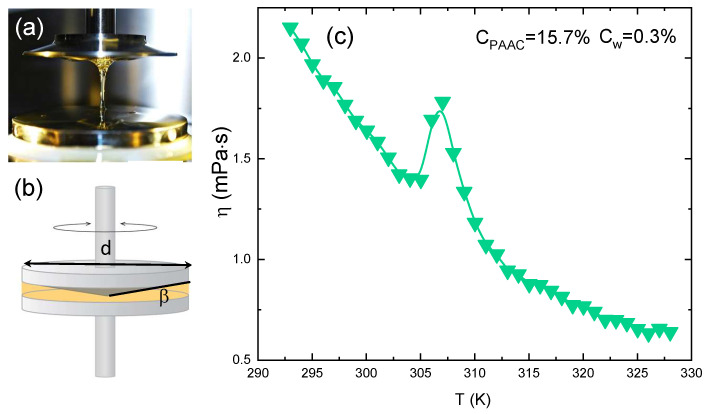
(**a**) Picture of the cone-plate geometry used to perform rheological measurements. (**b**) Schematic rapresentation of the used geometry (**c**) Viscosity of IPN microgels with $C_{PAAc}$ = 15.7% as a function of temperature at $C_{w}$ = 0.3%, pH = 5.5 and $\dot{\gamma }$ = 10 s${}^{-1}$.

**Figure 8 polymers-13-01353-f008:**
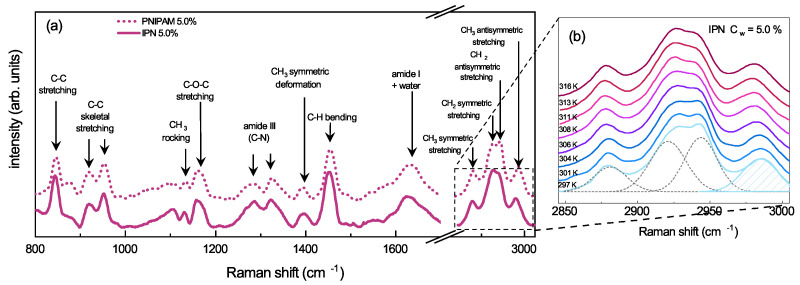
(**a**) Raman spectra at T = 316 K and C${}_{w}$ = 5.0% for PNIPAM and IPN microgels with C${}_{PAAc}$ = 19.6% and pH = 5.5. (**b**) Magnification for IPN microgels of the spectral range 2850 cm${}^{-1}$÷ 3000 cm${}^{-1}$. Four Gaussian contributions are reported as discussed in the text.

**Figure 9 polymers-13-01353-f009:**
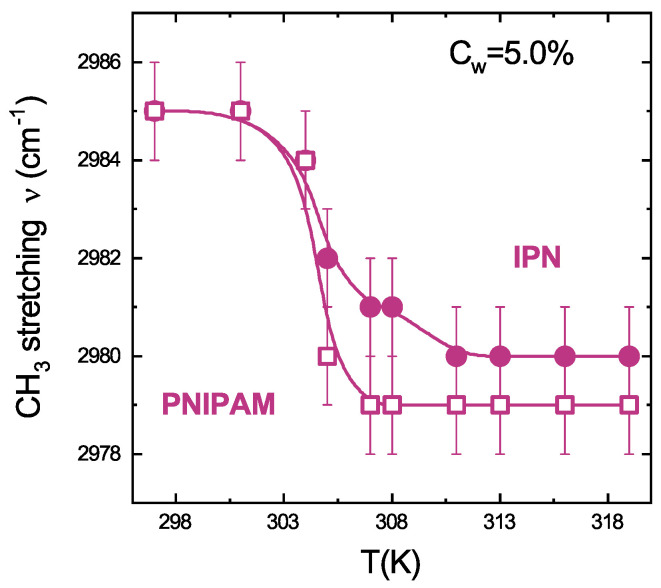
CH${}_{3}$ frequency as a function of temperature for PNIPAM and IPN microgels at C${}_{PAAc}$ = 19.2%, $C_{w}$ = 5.0% and pH = 5.5. Solid lines are guides to eyes.

**Figure 10 polymers-13-01353-f010:**
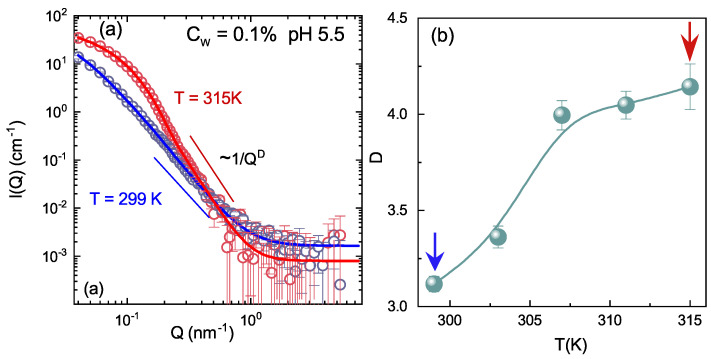
(**a**) SANS scattered intensity for IPN microgels at C${}_{PAAc}$ = 13.6% in D${}_{2}$O at $C_{w}$ = 0.1% and pH 5.5 for two temperatures: *T* = 299 K and *T* = 311 K, respectively, below and above the VPTT. Full lines are fits according to Equation (Equation 3). (**b**) Porod exponent, D, as a function of temperature as obtained through fits with Equation (Equation 3).

**Figure 11 polymers-13-01353-f011:**
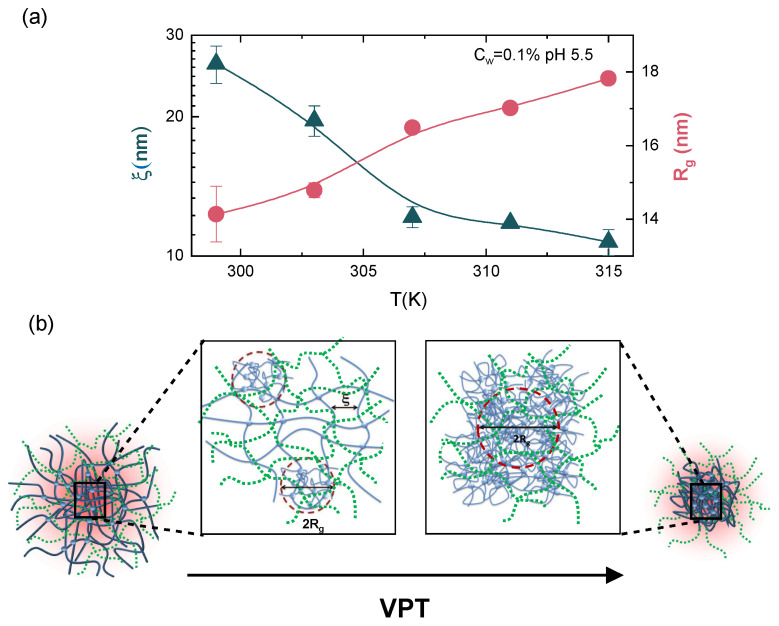
(**a**) Correlation length ξ (triangles) and gyration radius $R_{g}$ (circles) of the correlation domains as a function of temperature for IPN microgels at $C_{PAAc}$ = 13.6%, $C_{w}$ = 0.1% and pH 5.5. Full lines are guide to eyes. (**b**) Drawing of the local internal structure of IPN microgel particles below (left side) and above (right side) the VPT. The solid and dashed lines represent the two interpenetrating networks with average mesh size ξ. The dashed red circles in the left side panel (swollen state) evidence the regions of quenched inhomogeneities of average size (radius) $R_{g}$, due to the presence of cross-links. Increasing temperature, the collapse of the polymer networks induces a transition from an inhomogeneous structure (left side panel) to a porous solid-like structure (right side panel).

**Figure 12 polymers-13-01353-f012:**
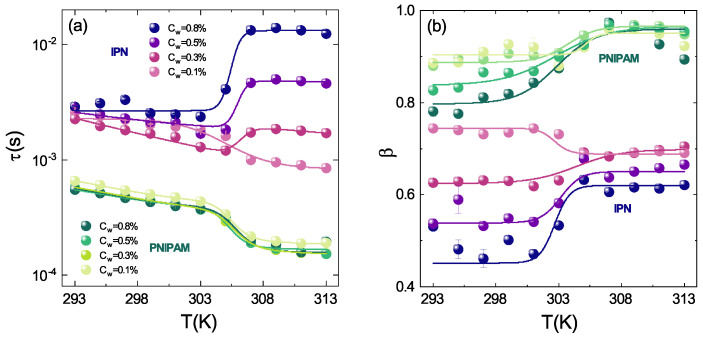
(**a**) Structural relaxation time and (**b**) shape parameter as a function of temperature for PNIPAM and IPN microgels at the indicated weight concentrations, at $C_{PAAc}$ = 19.2%, pH = 5.5 and Q = 0.018 nm${}^{-1}$. Solid lines are guides to eyes.

**Figure 13 polymers-13-01353-f013:**
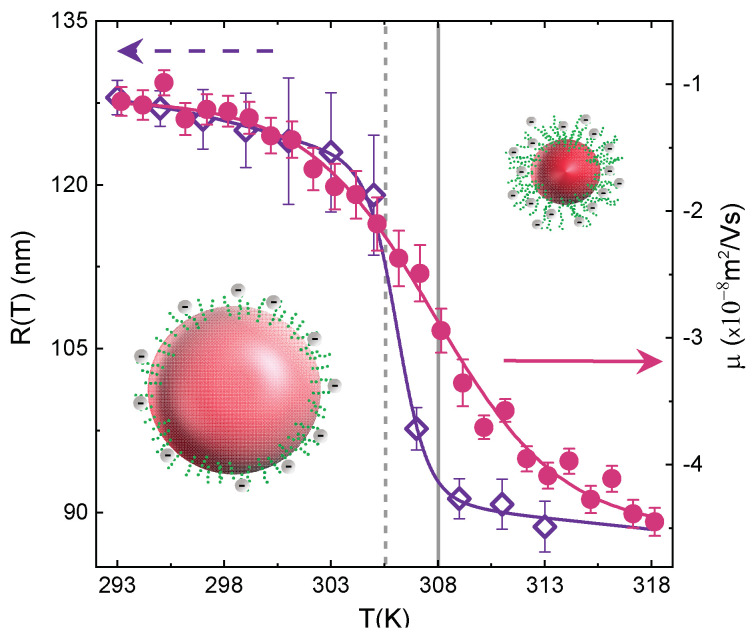
Comparison between electrophoretic mobility and hydrodynamic radius for IPN microgel at C${}_{PAAc}$ = 19.2%, low weight concentrations ($C_{w}$ = 0.05% for electrophoretic mobility and $C_{w}$ = 0.01% for DLS measurements) and pH = 5.5. Vertical dashed lines represent the VPT temperature ≈ 305 K and the electrophoretic transition temperature T${}_{0}$≈ 308 K.

**Figure 14 polymers-13-01353-f014:**
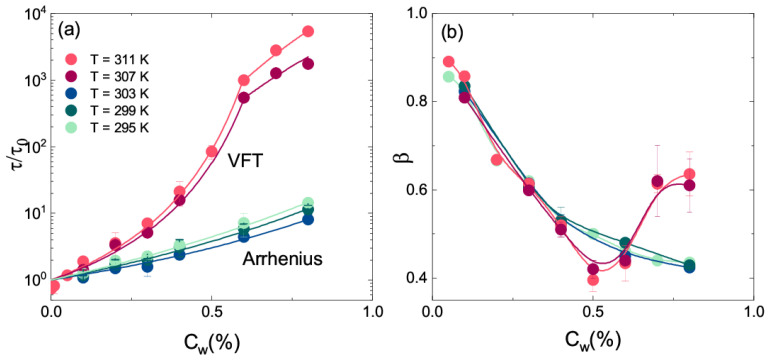
(**a**) Structural relaxation time τ and (**b**) shape parameter β from Equation (Equation 1) as a function of weight concentration for IPN microgel suspensions at the indicated temperatures, at $C_{PAAc}$ = 24.6%, Q = 0.018 nm${}^{-1}$ and pH = 5.5. Full lines represent the best fits with Equation (Equation 4) below the VPT and Equation (Equation 5) plus a power law at T> VPTT for τ and guides to eyes for β.

**Figure 15 polymers-13-01353-f015:**
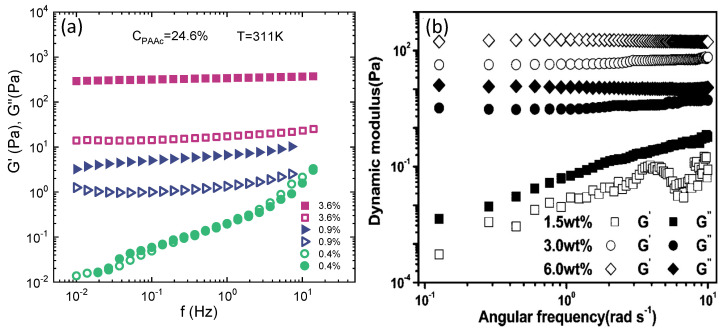
(**a**) Storage modulus $G^{\prime }$ (closed symbols) and loss modulus $G^{\prime \prime }$ (open symbols) versus frequency at the indicated concentrations for IPN microgels at $C_{PAAc}$ = 24.6% and T = 311 K. (**b**) Storage modulus, $G^{\prime }$ (open symbols) and loss modulus $G^{\prime \prime }$ (closed symbols) versus frequency at the indicated concentrations for IPN microgels at $C_{PAAc}$ = 16.7% and T = 310 K (figure adapted with permission from Ref. [59], Copyright 2021 American Chemical Society.)

**Figure 16 polymers-13-01353-f016:**
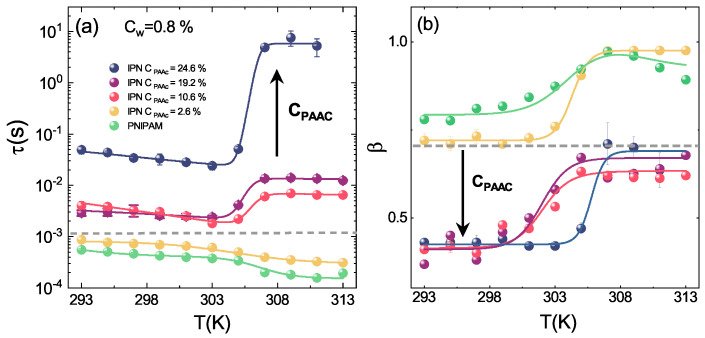
Temperature behaviour of (**a**) structural relaxation time and (**b**) β parameter for PNIPAM and IPN microgels at the indicated PAAc contents, at $C_{w}$ = 0.8%, pH 5.5 and Q = 0.018 nm${}^{-1}$. Solid lines are guides to eyes.

**Figure 17 polymers-13-01353-f017:**
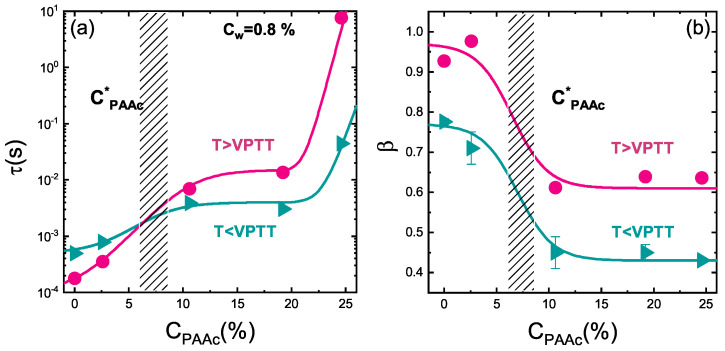
(**a**) Structural relaxation time and (**b**) β parameter as a function of the PAAc content at $C_{w}$ = 0.8% and pH = 5.5 below (T = 295 K) and above (T = 311 K) the VPTT. Solid lines are guides to eyes.

**Figure 18 polymers-13-01353-f018:**
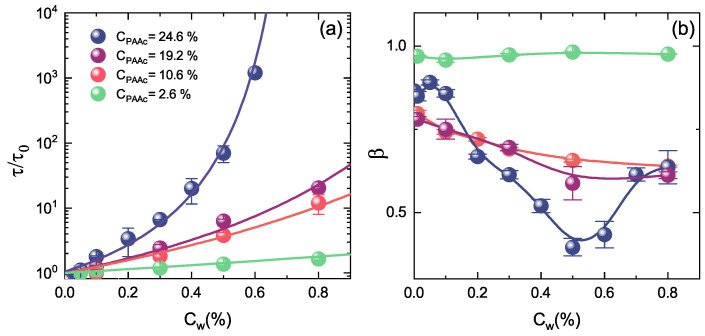
(**a**) Normalised structural relaxation time and (**b**) β parameter as a function of weight concentration at the indicated PAAc contents at *T* = 311 K, pH = 5.5 and Q = 0.018 nm${}^{-1}$. Solid lines are fits through Equation (Equation 5) for τ and guide to eyes for β.

**Figure 19 polymers-13-01353-f019:**
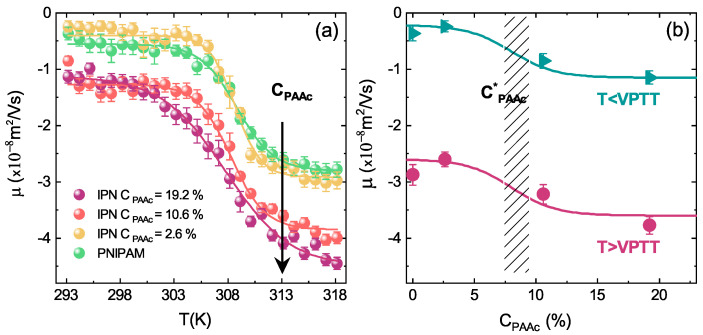
(**a**) Electrophoretic mobility as a function of temperature for PNIPAM and IPN microgels at the indicated PAAc content at low weight concentrations and pH = 5.5. Full lines are fits through the sigmoidal function as reported in [73]. (**b**) Electrophoretic mobility as a function of the PAAc content at low weight concentrations, at pH = 5.5 below (T = 295 K) and above (T = 311 K) the VPTT. Solid lines are guides to eyes.

**Figure 20 polymers-13-01353-f020:**
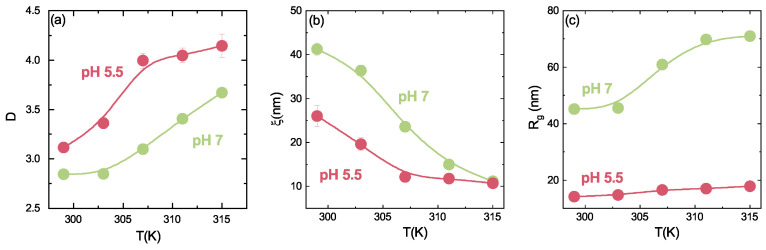
(**a**) Porod exponent D, (**b**) correlation length ξ and (**c**) gyration radius $R_{g}$ of the correlation domains of IPN microgels with C${}_{PAAc}$ = 13.6% as a function of temperature at fixed concentration $C_{w}$ = 0.1% and at acidic and neutral pH. Full lines are guide to eyes.

**Figure 21 polymers-13-01353-f021:**
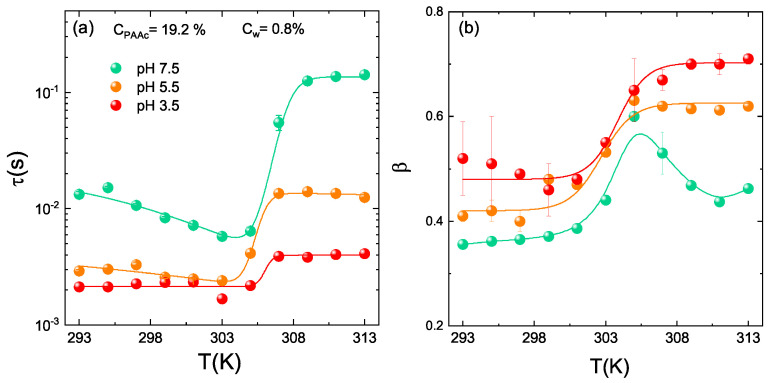
Temperature behaviour of (**a**) structural relaxation time and (**b**) β parameter for IPN microgels at C${}_{w}$ = 0.8% C${}_{PAAc}$ = 19.2%, at pH = 3.5, pH = 5.5 and pH = 7.5 at Q = 0.018 nm${}^{-1}$. Solid lines are guides to eyes.

**Table 1 polymers-13-01353-t001:** Composition of the prepared IPN microgels as assessed by combined ${}^{1}$H-NMR and elemental analyses.

	PNIPAM	PAAc	BIS
IPN C${}_{PAAc}$ = 2.6%	94.4	2.6	3.0
IPN C${}_{PAAc}$ = 10.6%	85.7	10.6	3.7
IPN C${}_{PAAc}$ = 15.7%	76.7	15.7	7.6
IPN C${}_{PAAc}$ = 19.2%	73.6	19.2	7.2
IPN C${}_{PAAc}$ = 24.6%	67.7	24.6	7.7

## Data Availability

The data that support the findings of this study are available from the corresponding author.

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
