# Peer review of "Chemical-Physical Behaviour of Microgels Made of Interpenetrating Polymer Networks of PNIPAM and Poly(acrylic Acid)"

_polymers, 2021, doi:10.3390/polym13091353_

Round 1

Reviewer 1 Report

The manuscript was prepared so poorly. The introduction part was inappropriate where the authors did not even know why they are designing this interpenetrating hydrogel. There is not enough literature survey and need for this material for a specific application. Additionally, there have been similar articles related to hydrogels of PNIPPAAM and Poly (acrylic acid) ( example: 

  • 10.1163/092050610X538722
  • DOI: 10.1007/s00396-014-3314-9

these are few examples. Therefore, I request to reject this manuscript in the current form.

Reviewer 2 Report

In this manuscript, the author studied the behavior of pNIPAm microgel that interpenetrating with PAA polymers through multiple technologies. Based on the results, I think it can be accepted for publication after addressing the following questions.

  1. The concentration of PAAc is based on the feeding ratio, could you please provide some information that shows the amount of PAAc inside the microgels?
  2. How stable of PAAc polymer inside the microgels? Since we know that microgels are highly porous and polymer chains can freely penetrate. 
  3. The author should provide explanation of difference between this microgel system vs pNIPAm-co-AAc microgels.

Reviewer 3 Report

The authors report on a very interesting article related to the “Chemical-physical behaviour of microgels made of interpenetrating polymer networks of PNIPAM and poly(acrylic acid)”.

The work is not very much innovative; however highlighting and elucidating underlying mechanisms in widely used temperature and pH responsive polymeric microgels.

Some suggestions below that should be included/ revised:

  1. Please in abstract, abbreviate once some terminology i.e. Dynamic light scattering as “DLS” and use it in the abbreviated form throughout the whole manuscript.
  2. In the introduction section (first sentences); please for the related literature split the references of the microgels especially PNiPAM and PAAc in different sectors i.e. medical [], optoelectronic [], sensing [], etc. applications. Which are the industrial applications the authors include [7-12] six references.
  3. Moreover, the introduction section should be extended to 6-7 paragraphs; i.e. each paragraph summarizing specific use of PNiPAM and/ or PAAc microgels and the mechanisms of the responsiveness how they endow the unique properties for such colloidal systems.
  4. Fig 1 should appear below the Experimental description corresponding paragraph. Fig 2 should also be placed below the 2.2. paragraph.
  5. Fig 3 should be moved to the “Results and Discussion” part- section.
  6. Please the figure placement within the document should be checked. Also Figure 19 appears in the Conclusions section.

In terms of originality, importance & scientific quality, relevance & contribution to the field and presentation, this manuscript is of good level. The synthesised systems do not exhibit any novelty; however the analysis of the microgel physicochemical behavior is thorough and excellent.

The manuscript should be improved in the points that have been indicated in order to make it more comprehensive while improving the quality overall.

The manuscript and its content are sufficiently novel to warrant its publication, however, after including and considering the minor additions and clarifications proposed.

Round 2

Reviewer 1 Report

  • I think the authors have made the research article more like a "Review article". Please be noted that both are different and I kindly request the authors to cut down the introduction and only highlight the points and necessity of your research into less than 3-4 pages. 
  • Additionally, there are too many figures and I do not think all of them needed to show in the main text only when they are necessary. So Accordingly please re-arrange the Figures in the revised manuscript.
  • The whole idea of any manuscript (especially research article) should be crisp and clear that the readers can understand easily.